# Supplemental Oxygen Alters the Airway Microbiome in Cystic Fibrosis

Jacob Vieira,[a] Sirus Jesudasen,[b] Lindsay Bringhurst,[a] Hui-Yu Sui,[a] Lauren McIver,[c] Katrine Whiteson,[d] Kurt Hanselmann,[e] George A. O'Toole,[f] Christopher J. Richards,[a] Leonard Sicilian,[a] Isabel Neuringer,[a] Peggy S. Lai[a]

aDivision of Pulmonary and Critical Care Medicine, Massachusetts General Hospital, Boston, Massachusetts, USA
bDepartment of Medicine, Massachusetts General Hospital, Boston, Massachusetts, USA
cDepartment of Biostatistics, Harvard T. H. Chan School of Public Health, Boston, Massachusetts, USA
dDepartment of Molecular Biology & Biochemistry, University of California, Irvine, California, USA
eSwiss I-Research and Teaching Institute, Zürich, Switzerland
fDepartment of Microbiology & Immunology, Geisel School of Medicine at Dartmouth, Hanover, New Hampshire, USA

Jacob Vieira and Sirus Jesudasen contributed equally to this work and should be considered co-first authors. Author order was determined with a random number generator.

**ABSTRACT** Features of the airway microbiome in persons with cystic fibrosis (pwCF) are correlated with disease progression. Microbes have traditionally been classified for their ability to tolerate oxygen. It is unknown whether supplemental oxygen, a common medical intervention, affects the airway microbiome of pwCF. We hypothesized that hyperoxia significantly impacts the pulmonary microbiome in cystic fibrosis. In this study, we cultured spontaneously expectorated sputum from pwCF in artificial sputum medium under 21%, 50%, and 100% oxygen conditions using a previously validated model system that recapitulates microbial community composition in uncultured sputum. Culture aliquots taken at 24, 48, and 72 h, along with uncultured sputum, underwent shotgun metagenomic sequencing with absolute abundance values obtained with the use of spike-in bacteria. Raw sequencing files were processed using the bioBakery pipeline to determine changes in taxonomy, predicted function, antimicrobial resistance genes, and mobile genetic elements. Hyperoxia reduced absolute microbial load, species richness, and diversity. Hyperoxia reduced absolute abundance of specific microbes, including facultative anaerobes such as *Rothia* and some *Streptococcus* species, with minimal impact on canonical CF pathogens such as *Pseudomonas aeruginosa* and *Staphylococcus aureus*. The effect size of hyperoxia on predicted functional pathways was stronger than that on taxonomy. Large changes in microbial cooccurrence networks were noted. Hyperoxia exposure perturbs airway microbial communities in a manner well tolerated by key pathogens. Supplemental oxygen use may enable the growth of lung pathogens and should be further studied in the clinical setting.

**IMPORTANCE** The airway microbiome in persons with cystic fibrosis (pwCF) is correlated with lung function and disease severity. Supplemental oxygen use is common in more advanced CF, yet its role in perturbing airway microbial communities is unknown. By culturing sputum samples from pwCF under normal and elevated oxygen conditions, we found that increased oxygen led to reduced total numbers and diversity of microbes, with relative sparing of common CF pathogens such as *Pseudomonas aeruginosa* and *Staphylococcus aureus*. Supplemental oxygen use may enable the growth of lung pathogens and should be further studied in the clinical setting.

**KEYWORDS** cystic fibrosis, persons with cystic fibrosis, pwCF, lung, microbiome, oxygen, hyperoxia, perturbation

Address correspondence to Peggy S. Lai, plai@mgh.harvard.edu.

The authors declare no conflict of interest.

**C**ystic fibrosis is a genetic disease arising from mutations in the cystic fibrosis transmembrane conductance regulator (CFTR) gene that causes defective chloride secretion. Loss of *CFTR* function leads to the production of thick, viscous mucus and poor pulmonary clearance that result in lifelong recurrent bouts of pulmonary infections. In the absence of proper channel function, there is often progressive lung function decline that may result in the need for supplemental oxygen and ultimately necessitate lung transplantation or cause death. Pulmonary infections in persons with cystic fibrosis (pwCF) are polymicrobial, with the cooccurrence of specific microbes in a community leading to altered antimicrobial susceptibility (1) and clinical severity (2). In the long run, a reduction in sputum microbial diversity has been associated with worsening lung function and disease progression corresponding to the observed succession of the microbiome toward dominance of pathogens such as *Stenotrophomonas* and especially *Pseudomonas* (3–6). With the introduction of *CFTR* channel modulator drugs such as ivacaftor, both lung function and sputum microbial diversity improve (7).

Historically, oxygen supplementation has been considered a benign medical intervention and is provided liberally in many clinical contexts. However, early evidence in animal studies alluded to a link between exposure to hyperoxia and adverse effects, including lung injury and reduced immune activity (8–10). Over time, this evidence has been corroborated in human observational studies and clinical trials, where supplemental oxygen targeting high patient oxygen levels increased the risk of bacteremia, ventilator-associated pneumonia, and higher mortality in critical illness (11–15). While proposed mechanisms for the deleterious effects of oxygen have focused on the generation of reactive oxygen species and injury to pulmonary cells, less is known about the effects of supplemental oxygen on the pulmonary microbiome (16, 17).

Supplemental oxygen is prescribed for approximately 11% of pwCF (18–23). In clinical studies of pwCF, use of supplemental oxygen therapy has been associated with more advanced disease (24, 25), although oxygen therapy has been traditionally viewed as simply a marker of disease severity rather than a contributor to disease progression (26). In this study, we tested our hypothesis that supplemental oxygen alters the airway microbiome in pwCF. Our approach used a previously validated method (27) for culturing sputum from pwCF in artificial sputum medium under various oxygen conditions. We performed taxonomic, predicted functional, antimicrobial resistance, and mobile genetic element profiling using shotgun metagenomic sequencing, with the use of spike-in bacteria to determine absolute microbial abundance.

## RESULTS

**Study population.** Study participants were recruited through the Massachusetts General Hospital Adult Cystic Fibrosis Center from November 2019 to March 2020. Characteristics of the 11 pwCF included in this study are described in Table 1. The average age was 29.2 years old. Six pwCF were on CFTR modulators, six were actively receiving antimicrobials, and one was on supplemental oxygen. Seven had impaired glucose tolerance. Eleven sputum samples, one from each of these pwCF, were obtained during routine outpatient clinic visits and underwent culture in artificial sputum medium (ASM) under 21%, 50%, and 100% oxygen. This yielded 110 samples that underwent shotgun metagenomic sequencing (Fig. 1). Of these, two samples as well as the negative reagent-only control failed library preparation and sequencing.

**Visual culture phenotypes.** Oxygen influenced the observed phenotypes of sputum cultures. At the time of inoculation, filter-sterilized artificial sputum medium is clear yellow in color. Figure S1 in the supplemental material contains photographs of microbial communities from the 11 pwCF after 72 h of culture under 21%, 50%, and 100% oxygen. These visual differences between the same original sputum sample cultured under different oxygen conditions were the first indication that hyperoxia alters airway microbial communities. In Fig. S1, panels A, C, F, and H show sputum cultures with uniform yellow-white turbidity without noticeable visual differences across oxygen conditions. Panel B has yellow-white turbidity but also has a ring-shaped pellicle at the liquid-air interface that grows more prominent with increasing oxygen

**TABLE 1** Characteristics of study participants[a]

| Characteristic | Value for participants[b] |
| --- | --- |
| n | 11 |
| Age in yr, median (IQR) | 29.2 (25.3, 39.7) |
| Male gender | 3 (27.3) |
| White race | 10 (90.9) |
| BMI, median (IQR) | 21.6 (18.9, 23.4) |
| Pancreatic insufficiency | 10 (90.9) |
| Condition = exacerbation | 4 (36.4) |
| Supplemental oxygen use | 1 (9.1) |
| Channel modulator | 6 (54.5) |
| Antibiotic use within past 90 days[c] | 9 (81.8) |
| Impaired glucose tolerance | 7 (63.6) |
| Insulin use | 4 (36.4) |
| % Predicted FEV1, median (IQR) | 0.63 (0.42, 0.86) |
| % Predicted FVC, median (IQR) | 0.77 (0.58, 1.00) |

[a]Summary of medical characteristics of the 11 study participants whose sputum was used for these experiments and the resulting analyses. IQR, interquartile range; BMI, body mass index; $FEV_1$, forced expiratory volume in 1 s; FVC, forced vital capacity. FEV1 and FVC are measures of quality of lung function.
[b]Values are number (%) unless indicated otherwise.
[c]6 (54.5%) participants on antibiotics at time of sputum collection.

concentration. Panel D has strong gray-black pigmentation at 21% oxygen which is entirely absent under 50% and 100% oxygen. Panel E also displays color change, with yellow-orange pigmentation at 100% oxygen. Panel G has large orange clumps of growth at 21% and 50% oxygen but not at 100% oxygen. Panel I contain numerous white growth clumps that are much more numerous at 21% and 50% oxygen than at 100% oxygen. Panel J shows drastically decreased turbidity at 100% oxygen, suggestive of reduced growth. Lastly, panel K also shows color differentiation, with green pigmentation at 21% oxygen that was not present under 50% and 100% oxygen.

**Metagenomic sequencing.** Uncultured sputum and aliquots of sputum cultured in ASM from the indicated oxygen and time culture conditions underwent nucleic acid extraction and metagenomic sequencing on the Illumina NovaSeq platform. After trimming of low-quality reads, removal of host reads, and removal of spike-in reads, a median of 23.3 (interquartile range, 10.8 to 33.5) million reads per sample (78.7%) remained. A total of 7.4% of reads failed quality control, 1.2% were human, and 12.6% mapped to spike-in control bacteria. The minimum observed final sample read count was 148,882 reads. Figure S2 shows that even rarefaction to this minimum observed read count produces full saturation of species detection, indicating that at the level of deep sequencing performed in our study, differences in sequencing depth between samples did not affect species diversity estimates. The bioBakery3 (28) suite of tools was used to generate profiles for microbial taxonomy, predicted pathways, antimicrobial resistance (AMR) genes, and mobile genetic elements (MGE). Figure S3 shows a comparison of microbial species detected in uncultured sputum and in sputum cultured at 21% oxygen for 48 h, annotated with known oxygen tolerance based on preexisting literature. Anaerobes were detected in both uncultured sputum and sputum cultured under 21% oxygen.

**Microbial load.** Absolute cellular counts for the overall community and for each species were estimated by the addition of the spike-in control bacterium (29, 30) *Imtechella halotolerans* to the sample at the time of nucleic acid extraction. *Imtechella halotolerans* is a halophile not found in human microbial communities, allowing us to calculate total microbial load and species-specific counts based on the resulting sequencing data. Higher oxygen levels decreased the absolute microbial load, while longer culture times increased the absolute microbial load (Fig. 2). For each 1% increase in oxygen above 21%, the $log_{10}$ cells per milliliter absolute microbial load estimate decreases by $3.55 \times 10^{-3}$. Culturing for 72 h under 100% oxygen compared to 21% oxygen for 72 h reduces estimated microbial load by half, from 2.60 billion cells per milliliter to 1.36 billion cells per milliliter.

**Microbial diversity.** Figure 2 outlines the effect of culture condition on the number of unique species per culture and the inverse Simpson and Shannon alpha diversity indices.

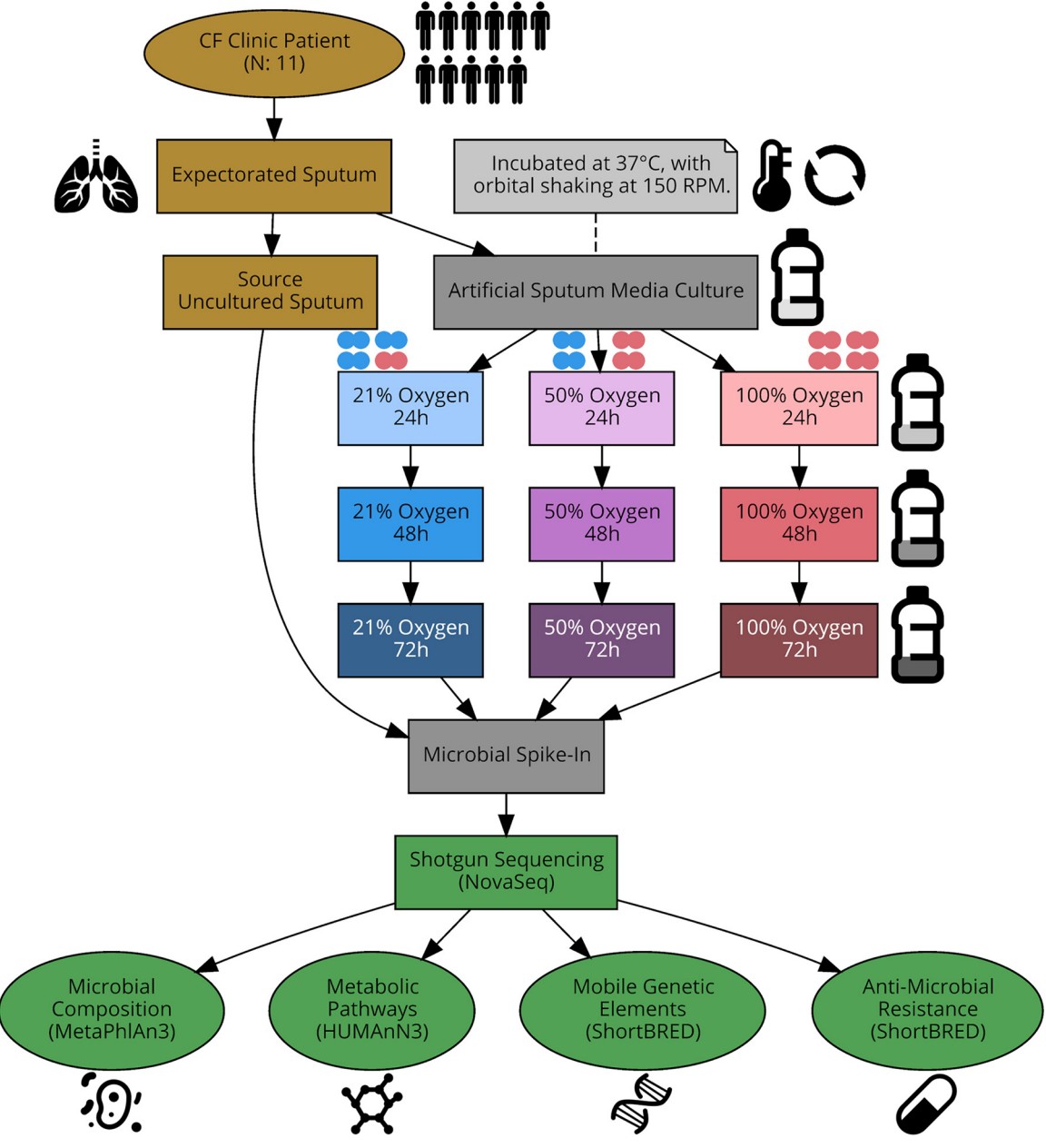

**FIG 1** Overview of study design. Each patient sputum sample generated 10 samples (9 cultured and 1 uncultured) for metagenomic sequencing. Sputum was cultured in artificial sputum medium under 21%, 50%, and 100% oxygen atmospheres, with aliquots taken at 24, 48, and 72 h. One aliquot was processed uncultured as the "source" sputum.

Hyperoxia decreased the alpha diversity of the airway microbial communities. Culture time conversely increases alpha diversity, though the observed number of species remained unchanged. The observed species count as well as the inverse Simpson and Shannon alpha diversity indices decrease as oxygen increases. For each 1% increase in oxygen above 21%, mixed-effects models predict that observed species decrease by $7.19 \times 10^{-2}$, inverse Simpson diversity decreases by $4.86 \times 10^{-3}$, and Shannon diversity decreases by $3.26 \times 10^{-3}$. Culturing the same starting community for 72 h under 100% oxygen compared to 21% oxygen reduces the number of unique species from 15.67 to 9.99 and reduces other measures of alpha diversity and community evenness. These estimates point to an overall community depletion effect from exposure to hyperoxia, where the diversity of microbes decrease as oxygen exposure increases.

To test the hypothesis that hyperoxia affected overall microbial community structure,

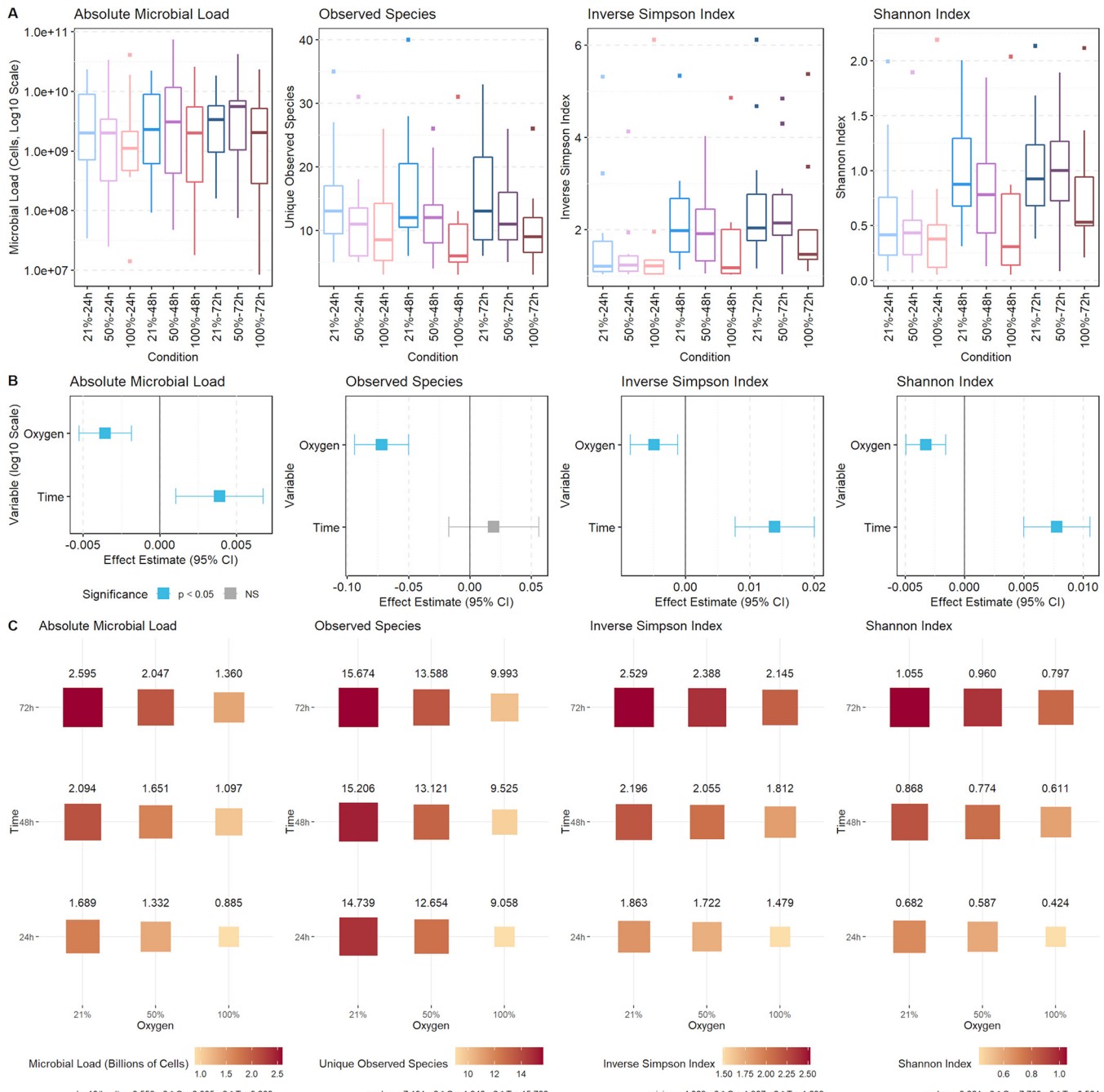

**FIG 2** Hyperoxia reduces microbial load and community diversity. (A) Distributions of microbial load and alpha diversity metrics stratified by culture condition. (B) Estimated effect size and 95% confidence intervals for the effect of oxygen and time on microbial load and alpha diversity from linear mixed-effects regressions. (C) Balloon plot with predicted values for microbial load and alpha diversity for each oxygen and time condition.

we performed beta diversity ordination with permutational analysis of variance modeling (Fig. S4). Both oxygen and time significantly impacted community composition. As seen in other airway microbiome studies from pwCF (31), there are very strong subject-specific effects, i.e., the airway microbiome of each pwCF is unique. Despite these strong subject-specific effects, oxygen remains a significant predictor of microbial community structure for both the Bray-Curtis dissimilarity ($R^2 = 0.01$, $P = 0.003$) and the Jaccard index ($R^2 = 0.01$, $P = 0.007$). Bray-Curtis, Jaccard, and nonmetric multidimensional scaling (NMDS) distances all revealed similar patterns of separation between samples.

Hyperoxia additionally influenced both the alpha and beta diversity of predicted

functional, antimicrobial resistance, and mobile genetic element profiles (Fig. S5). Similar to the patterns noted in the taxonomic profiles, oxygen decreases and time increases the observed alpha diversity of functional profiles. Using mixed-effects models, the same airway microbial community cultured for 72 h under 100% oxygen compared to 21% oxygen reduced the number of observed pathways from 307.7 to 268.5, that of mobile genetic elements from 16.2 to 12.5, and that of antimicrobial resistance genes from 51.9 to 44.3. The effect size of oxygen on microbial community structure was largest in predicted functional profiles (Bray-Curtis $R^2$ = 0.02, $P$ = 0.002), followed by taxonomy (Bray-Curtis $R^2$ = 0.01, $P$ = 0.003) and then predicted antimicrobial resistance genes (Bray-Curtis $R^2$ = 0.005, $P$ = 0.015).

Figure 3 depicts the microbial community profiles of cultured and uncultured sputum stratified by patient, showing both the relative and absolute abundances of microbial taxa. There were large differences in community composition between participants, indicating strong subject-specific effects. Most uncultured raw sputum had high relative and absolute abundances of *Staphylococcus aureus* and/or *Pseudomonas aeruginosa*.

**Differential effect of hyperoxia on microbial taxonomy and function.** Differential abundance testing was performed to evaluate the effect of oxygen on individual microbial species and predicted functional pathways. The differential effects of hyperoxia for all microbial taxa are available in Table S1, and those for all functional pathways are in Table S2. The pathways tested were subsequently manually curated to a subset of potential relevance to hyperoxia, including pathways related to fermentation, the electron transport chain, respiration, and metabolism. The differential effects data for this subset are available in Table S3. A complete list of inclusion and exclusion criteria for pathway curation are available in Table S4.

As noted earlier, oxygen overall reduced the absolute abundance of detected microbial species. The degree of this negative impact, however, varies widely by species (Fig. 4). Of the 10 most affected organisms, eight are obligate or facultative anaerobes, including *Rothia mucilaginosa*, *Actinomyces oris*, and multiple *Streptococcus* species. The two fungi among this set, *Candida albicans* and *Aspergillus fumigatus*, are both eradicated under 100% oxygen. Conversely, the most oxygen-tolerant species are aerobes or facultative anaerobes commonly classified as canonical pathogens in persons with cystic fibrosis, including *Pseudomonas aeruginosa*, *Staphylococcus aureus*, and *Stenotrophomonas maltophilia*.

To determine whether different microbial clades occupied similar functional niches, we examined associations between microbes and functional pathways using normalized Spearman's correlation and hierarchically clustered these taxonomy-function correlations in Fig. 5. Microbial taxa and pathways significantly altered by hyperoxia are annotated with blue boxes. Hierarchical clustering suggests the existence of five functional niches, each occupied by microbes performing similar functions in the community. *Pseudomonas aeruginosa* and *Serratia marcescens* form the backbone of the first cluster. *Klebsiella* species form their own tight cluster. *Staphylococcus* and *Burkholderia* species form a third cluster. A fourth group consists mostly of various facultative anaerobes, including *Rothia* and various *Streptococcus* species. The final group includes *Stenotrophomonas maltophilia* as well as the eukaryotes *Candida* and *Aspergillus*. These last two groups of species are most strongly impacted by hyperoxic conditions. Eukaryote-specific pathways were reduced by hyperoxia, which is associated with the large reduction in fungi and other eukaryotes under hyperoxic conditions. A wide variety of cellular functions are also impacted by the introduction of hyperoxia, including glycolysis, synthesis of electron transport carriers, nucleotide degradation, and fermentation, reflecting the alteration of the previously described groups.

**Effect of hyperoxia on microbial cooccurrence networks.** Although it is often assumed that microbial communities with higher diversity are also more stable, this is not always the case, as ecological models indicate that competitive relationships may stabilize microbial networks (32). Thus, we evaluated the effect of hyperoxia on microbial cooccurrence networks (Fig. 6) and compared network statistics for communities cultured under 21% and 100% oxygen. Exposure to a hyperoxic environment leads to global changes in network topology (Table S5). Comparing the overall similarities of the two networks yields an adjusted Rand index of 0.462 ($P$ < 0.001), indicating only 46.2% agreement in microbial

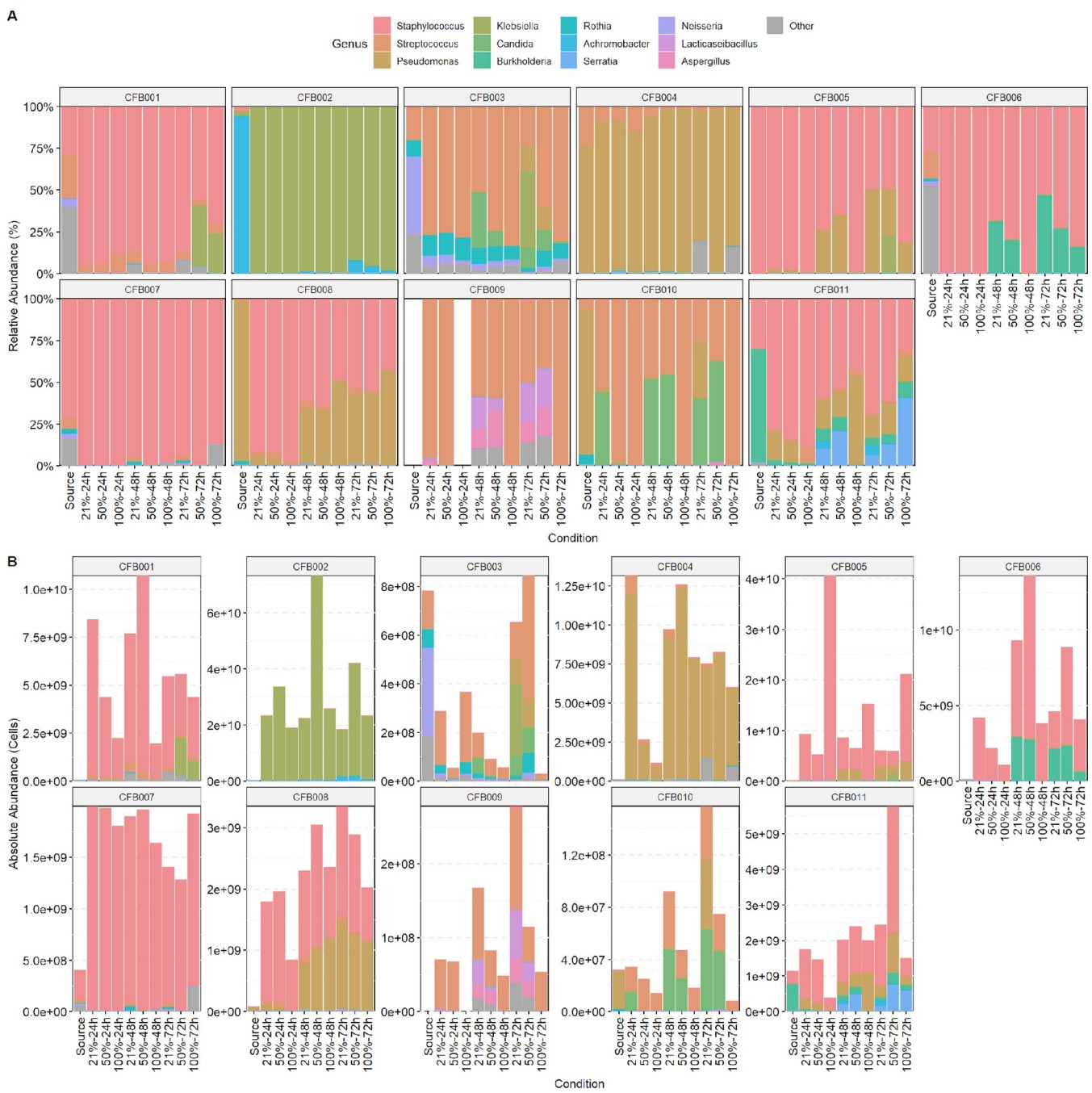

**FIG 3** Per-participant sputum community taxonomic profiles. Leftmost "source" bar corresponds to uncultured sputum samples; the remainder represent the nine culture conditions. Colors correspond to the 12 most abundant genera, with the remainder grouped in gray as "other." (A) Relative abundance community profiles, grouped by study participant. (B) Absolute abundance community profiles, grouped by study participant. Calculated using spike-in bacteria. The y axis differs between study participants due to differences in microbial load.

pair placement between the two sets. There is 92% dissimilarity between global degree centrality ($P = 0.004$) and a shift in network density from 0.308 to 0.150 ($P = 0.068$) under 100% oxygen conditions. Hyperoxia leads to fragmentation of the microbial network. While the normoxic microbial association network is unified into a single component, the hyperoxic network is broken into 16 components ($P = 0.001$), 12 of which are singlets isolated by the strong depletion of that species' presence under hyperoxia. These three metrics point to a significant overall sparsification of microbial associations under hyperoxic conditions. Within the remaining sparser network under hyperoxia, the cluster coefficient increases from 0.688 to 0.841 ($P = 0.002$), indicating tighter cluster formation among

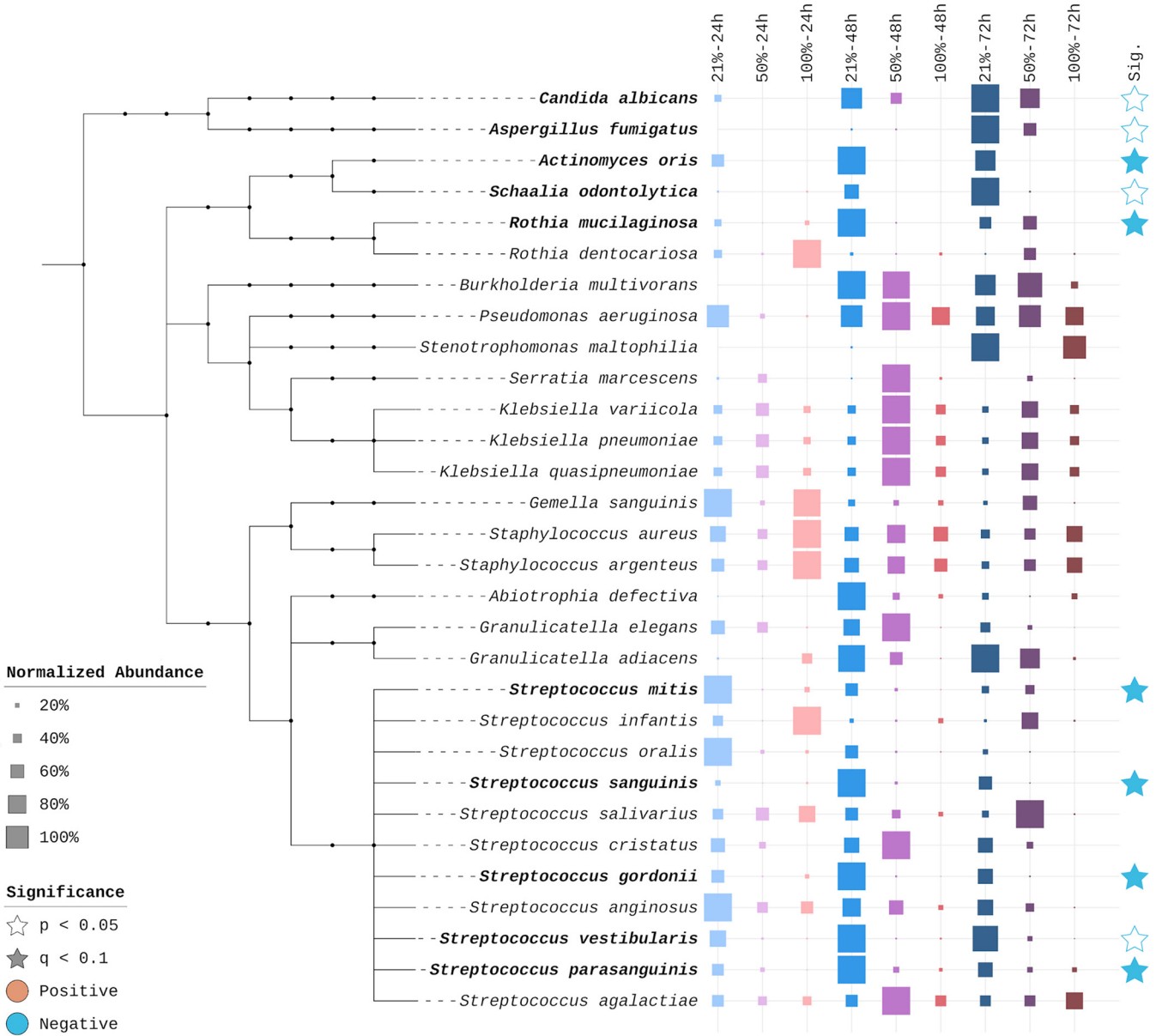

**FIG 4** Per-species differential effects of hyperoxia. Left, phylogenetic tree of the 30 most prevalent microbial species. Names in bold represent species identified as significantly impacted by hyperoxia. Middle, taxon-normalized conditional abundance values for each species. The highest conditional mean for each taxon is set to 100%. Right, significance level of differential effect of oxygen. Solid blue stars indicate $q$ values of <0.1 after Bonferroni-Hochberg multiple-hypothesis correction; white stars indicate raw $P$ values of <0.05 but $q$ values of >0.1.

the remaining relationships. Hyperoxic conditions may have reduced competition, approximated by the percentage of negative edges (negative correlations) between microbial species. The overall negative edge percentage decreased from 12.1% to 2.3% (21% versus 100% oxygen, $P = 0.004$), indicating a depletion of significant mediating (competitive) relationships. Global dissimilarity in eigenvector centrality is notably the weakest change, with only 38.1% dissimilarity ($P = 0.998$), suggesting that despite these changes, the most influential microbes in the network largely remain the same. A cluster of mostly facultative anaerobes such as *Streptococcus*, *Abiotrophia*, and *Actinomyces* (Fig. 6, colored in orange) is most perturbed by the increase in oxygen concentration.

## DISCUSSION

In this study, where we cultured sputum from pwCF in artificial sputum medium under normoxic and hyperoxic conditions, we demonstrate that supplemental oxygen

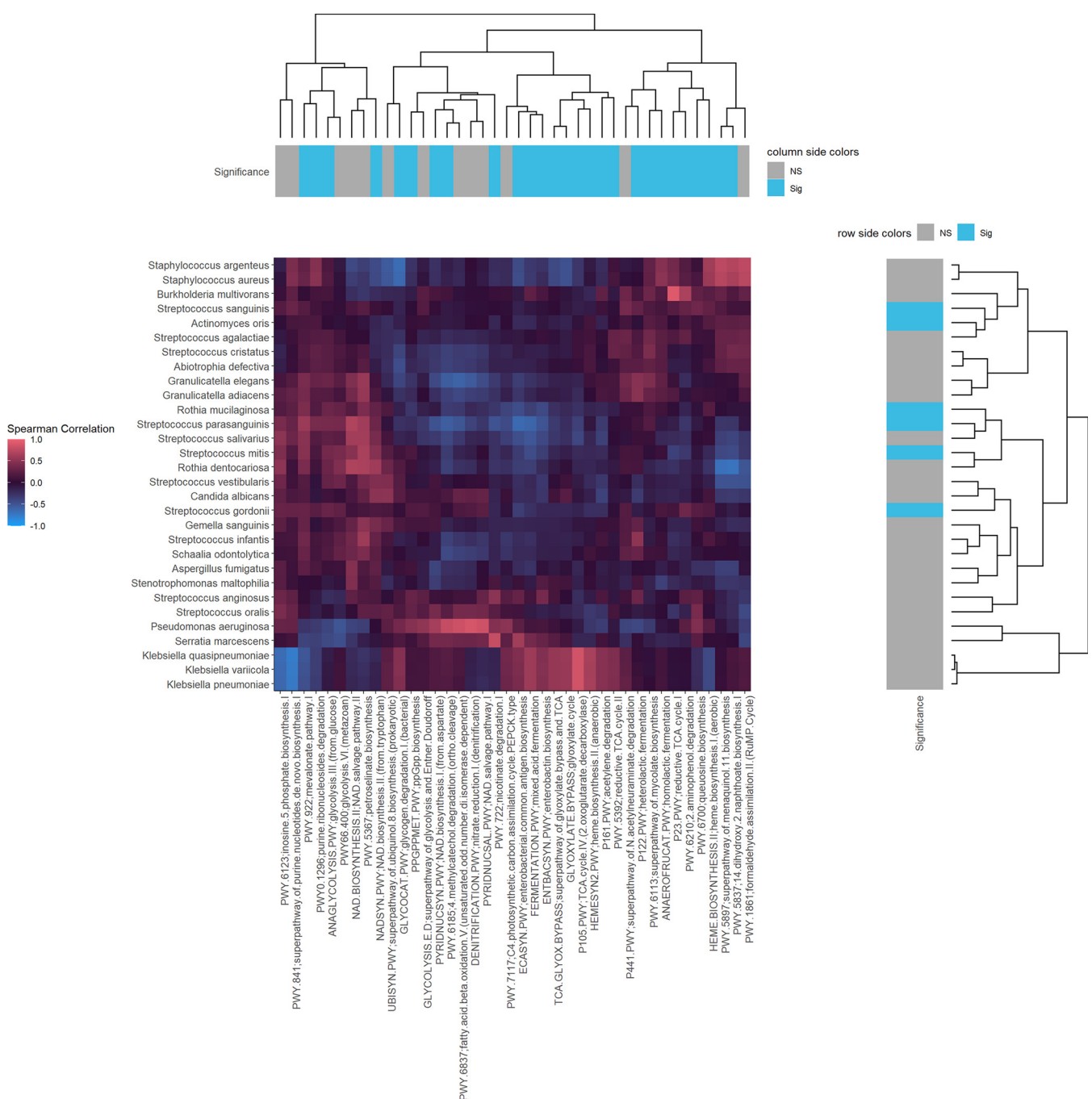

**FIG 5** Associations between predicted functional pathways and CF microbes. The y axis contains the 30 most prevalent microbial taxa, and the x axis contains a curated subset of predicted functional pathways, including fermentation, the electron transport chain, respiration, and metabolism pathways. Significance annotations indicate microbes/pathways that are significantly impacted by hyperoxia after Bonferroni-Hochberg multiple-hypothesis correction with a threshold q value of <0.1; blue boxes indicate q values of <0.1, gray boxes indicate q values of ≥0.1 The main heat map plots the Spearman rank correlation of each microbe against each of the selected pathways. Dendrograms relate microbes by functional pattern and functions by microbial pattern.

significantly alters airway microbial communities, with reduced absolute microbial load and reduced alpha diversity of microbial species, predicted microbial community function, and predicted antimicrobial resistance genes and mobile genetic elements. However, the effect of oxygen has a differential element, decreasing the absolute abundance of some facultative anaerobes such as *Rothia mucilaginosa*, *Streptococcus* species, and *Actinomyces oris* while having no significant effect on the absolute abundance of typical CF pathogens such as *Pseudomonas aeruginosa* and *Staphylococcus aureus*. The influence of supplemental oxygen is greater at the functional level than at the

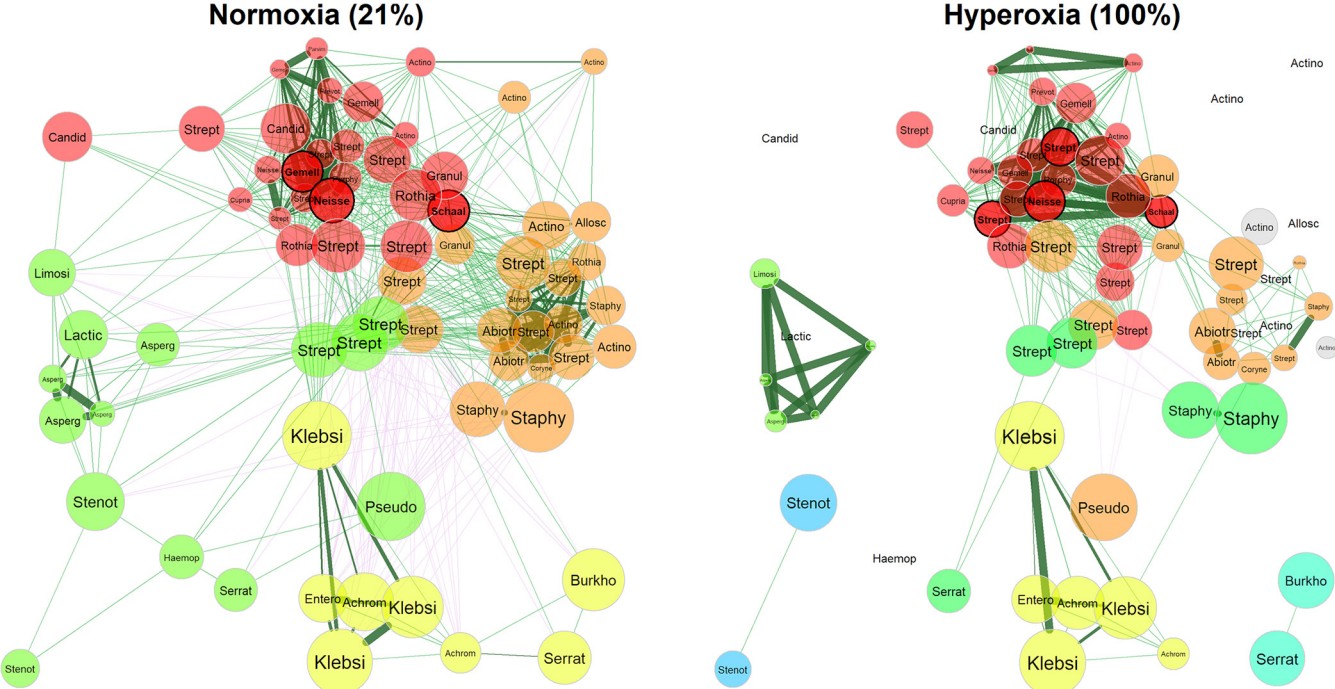

**FIG 6** Hyperoxia alters microbial network topology. Comparison of microbial association networks from sputum cultured under 21% and 100% oxygen. Network associations were calculated based on Spearman's correlation from absolute abundance data and sparsified via adaptive Bonferroni-Hochberg corrected *t* tests with a *q* value threshold of 0.1. Edge weights were based on Spearman's correlation, with synergistic (positive) relationships indicated by green lines and mediating (negative) relationships indicated by magenta lines. Each node corresponds to a bacterial or fungal species, with node color determined by association cluster and node size determined by average absolute abundance scaled using a $\log_{10}$ transformation.

taxonomic level. These findings support our hypothesis that the use of supplemental oxygen as a therapy may have unintended consequences on the airway microbiomes of pwCF. The decrease in alpha diversity due to supplemental oxygen may be analogous to the effect of antimicrobial administration and, importantly, raises the possibility that oxygen may enable the growth of pathogens. This possibility should be verified in future clinical studies.

These results parallel those of other studies examining questions of potential impacts of hyperoxia on airway microbiota. For example, in a study utilizing amplicon sequencing of the 16S rRNA gene on lung homogenate in a mouse model of acute lung injury, hyperoxia altered the lung microbiome in both newborn and adult mice (33). In this study, there was a decrease in the relative abundance of anaerobes such as *Clostridia* and *Bacteroidia* and a corresponding increase in the relative abundance of oxygen-tolerant microbes such as *Staphylococcus*. Changes in lung microbial communities due to oxygen preceded the development of acute lung injury, and germfree mice were protected from oxygen-induced acute lung injury, indicating that the deleterious effect of hyperoxia on lung injury is at least partially mediated by the lung microbial community (33). Another study examined the effect of oxygen gradients on microbial communities using the WinCF model, where sputum from pwCF was cultured in capillary tubes under 21% oxygen to mimic plugged airways (34). Steep oxygen gradients ranging from normal to low oxygen levels formed naturally based on the distance to the air-liquid interface. This oxygen gradient split the airway community into separate communities of oxygen-tolerant pathogens and anaerobes (34). Although this study focused on normoxic to hypoxic conditions while ours focused on hyperoxic conditions, there was strong parity between the affected taxa in this study and our findings, including high growth performance by *Pseudomonas* under normal or elevated oxygen conditions and the reduction of *Actinomyces*, *Prevotella*, and *Streptococcus*.

Anaerobic bacteria are an important component of the airway microbiome in pwCF (35) and may be associated with improved outcomes (36). The climax-attack model (37) for

airway microbial communities of pwCF hypothesizes the existence of two major functional communities: the attack community, which induces strong innate immune responses typically seen in exacerbations of cystic fibrosis, and a climax community associated with slower-growing communities not associated with exacerbations. Anaerobes such as *Rothia* have been implicated as part of the stable climax microbial community (37). In one study, pulmonary exacerbations and an attack community correlate to a diminished relative abundance of *Rothia*, with resolution of the exacerbation associated with the reemergence of *Rothia* and a return to a climax community (38). A culture-based study found that higher colony counts of anaerobic bacteria on sputum cultures were associated with a better lung clearance index and lower systemic inflammatory markers (39). The presence of certain anaerobic bacteria such as *Veillonella* in the airway microbiome of pwCF has been associated with better lung function (40). During pulmonary exacerbations of cystic fibrosis, the abundance of anaerobes such as *Streptococcus sanguinis*, *Prevotella melaninogenica*, and *Porphyromonas catoniae* decrease, suggesting that decreasing abundance of anaerobes is associated with exacerbations (41), although in this study, a higher abundance of *Veillonella parvula*, another anaerobe, was observed during pulmonary exacerbations. Anaerobes, however, produce short chain fatty acids that have been associated with increased inflammatory responses in cell culture studies of airway epithelia (42, 43), may produce fermentation products that support the growth of *Pseudomonas aeruginosa* (44), and may further contribute to antimicrobial resistance in recognized CF pathogens, including *Pseudomonas* (45). Thus, there is overall some debate as to whether an increased representation of anaerobes in the airway microbiome of pwCF is overall beneficial for health. Regardless, our study shows that oxygen supplementation in pwCF may decrease the absolute abundance of anaerobes and facultative anaerobes that are important constituents of the airway microbiome in pwCF and may restrain the growth of canonical lung pathogens. In the context of the climax-attack model, stable microbial communities have been associated with mild disease, while shifting unstable communities are associated with more severe disease outcomes (46).

The role of the pulmonary microbiome in health may run parallel to that of the gut microbiome, where alterations to microbial composition and diversity may lead to subsequent consequences in terms of clinical health. Decreased gut microbial biodiversity following antibiotic use has been linked to an increased susceptibility to opportunistic infection (47). In CF, thickened mucus leads to areas of relative hypoxia throughout the lungs and subsequent enrichment of anaerobes (48, 49). Supplemental oxygen can potentially alter oxygen gradients within the lungs, which may lead to the depletion of the anaerobes observed with reduced absolute microbial growth and biodiversity under conditions of hyperoxia. Interestingly, we found that the pathogen *S. aureus* appeared to be the most oxygen-tolerant microbe in our study. It is unclear whether the relative dominance of *S. aureus* with hyperoxia in the lungs may promote accelerated growth to fill the ecological niche of other depleted species, as this may bestow an associated risk of pulmonary infection. While *S. aureus* is a common colonizer in early CF, its presence as a colonizer later in CF disease has not been associated with alterations in lung function and thus the long-term clinical implications need further study (50).

Excessive oxygen has been associated with worsened clinical outcomes in mortality and infection risk in critical illness. Our work provides evidence of potential unintended consequences of supplemental oxygen use with alterations in the airway microbiome of pwCF. At present, clinical efforts limiting oxygen use have focused on conservative oxygen targets when administering supplemental oxygen in order to minimize excessive use. Oxygen supplementation may disrupt the airway microbiome by promoting growth of canonical CF pathogens such as *Pseudomonas* and *S. aureus* while depleting anaerobes and fungi. Thus, it may be prudent for clinicians to monitor sputum microbial communities following initiation of supplemental oxygen for emergence of harmful pathogens. The impact of hyperoxia on microbial growth in our study raises the possibility that supplemental oxygen may impact other clinically relevant outcomes,

such as antimicrobial susceptibility (51). While we were underpowered to explore such effect modification in this study, exploration of the effects of supplemental oxygen over a longer time horizon, or trending changes in airway microbiology from pwCF following oxygen initiation, would provide further insight on additional potential risks and benefits of oxygen use. Oxygen is often initiated at the time of clinical deterioration, when other therapies such as antimicrobials may be initiated, making it difficult to disentangle the confounding effects of these therapies from oxygen alone in an observational study.

This study had several strengths. We evaluated the effect of supplemental oxygen using an ecological approach, where we cultured airway microbial communities using sputum from pwCF rather than single isolates. It is well known that canonical CF pathogens such as *Pseudomonas aeruginosa* exhibit a different phenotype when studied in isolation than when studied in the presence of a broader microbial community (52, 53). We used artificial sputum formulated to mimic the composition of sputum for pwCF (27) rather than using rich medium more typically used in clinical microbiology laboratories. We have previously shown that sputum cultured under the 21% oxygen condition in our model recapitulates microbial communities of uncultured sputum when assessed by metagenomic sequencing (27). We evaluated changes in absolute rather than relative abundance in our taxonomic profiles with the use of spike-in controls prior to nucleic acid extraction, although we did not verify the derived absolute abundance estimates. Absolute rather than relative abundance may be an important determinant of clinical outcome (54) and does not have inherent statistical limitations associated with compositional data sets (55). While most existing studies on the airway microbiome in pwCF have leveraged amplicon sequencing, we used deep metagenomic sequencing, thus allowing for evaluation of all microbial domains, finer taxonomic resolution to the species level, and addressing not just microbial taxonomy but also potential function of the airway microbial community.

There are limitations to this study approach that must be considered in evaluating these findings. First, we employed an *ex vivo* model for culture that we have previously shown can recapitulate uncultured airway microbial communities from pwCF (27). However, while useful, all models have inherent limitations, and the exact effects *in vivo* may differ somewhat from those observed in our *ex vivo* approach. Our model does not account for host mechanisms to reduce microbial load, which include cough, other mucus clearance mechanisms, and the host immune system, as well as host-derived microbial nutrients. While we do not know the role that these host factors may play in the response to hyperoxia, it is of relevance that in human studies of ventilator-associated pneumonia where patients require life support for respiratory failure and therefore high levels of supplemental oxygen, the two most common pathogens detected are *Pseudomonas aeruginosa* and *Staphylococcus aureus*, two of the most oxygen-tolerant species identified in our study (56). Thus, despite inherent limitations of all model systems (57), the results of our study may generalize to the clinical setting, although this will need to be further verified in future studies.

In this model system, we did not include a purely anaerobic condition, which is a limitation, although we did still detect anaerobes in our 21% oxygen culture condition. Steep oxygen gradients naturally form in microbial communities in many environments, including both *in vivo* in humans and in experimental laboratory conditions. While in health the airway lumen is an aerobic environment, bacterial respiration at the air-liquid interface in human lung environments rapidly consumes oxygen, thus allowing anaerobic bacteria to grow below the air-liquid interface. Oxygen concentration gradients proportional to the distance from the surface of airway mucus have been documented in the airways of persons with established CF (58, 59). Measurements taken in CF sputum samples show that sputum samples have steep oxygen gradients, with little oxygen measured just 1 mm below the surface of the sputum sample (58). In experimental models, such as the WinCF model that mimics one form of mucus aggregation (complete plugging of small airways) seen in pwCF, anaerobic conditions naturally develop below the air-liquid interface even

though cultures are incubated under normoxic (21% oxygen) conditions (34, 60). Even in our aerobic culture model at 21% oxygen, a variety of anaerobes and facultative anaerobes were still identified, including *Actinomyces*, *Veillonella*, *Gemella*, and others (see Fig. S3 in the supplemental material). With the advent of CFTR channel modulators which potentiate CFTR function and thus normalize mucus viscosity and mucus transport (61), the prevalence of complete plugging of the small airways and alveoli is likely to decrease, making anaerobic conditions in the airways of pwCF less frequent, although it will likely always remain relevant due to the presence of an air-liquid interface in the lungs.

Our airway samples were obtained through spontaneous expectoration rather than bronchoscopy. While expectorated sputum raises the possibility of oral contamination, sputum has been validated as an accurate measure of the airway microbiome in pwCF with strong similarity to lower respiratory tract samples (62). Finally, our sample size at the patient level was small due to the large number of samples generated for deep sequencing in order to test different time and oxygen conditions. While we clearly detected differences in omnibus measures of the microbiome such as alpha and beta diversity, a larger sample size may have increased our power to detect changes in abundances of specific microbes and allowed for exploration of effect modification between antibiotic use and hyperoxia or between channel modulator use and hyperoxia.

Our study demonstrates that in this validated model system, hyperoxia alters the airway microbiome in pwCF and therefore may have unintended effects in reducing airway microbial diversity by depleting less-oxygen-tolerant species, including anaerobes. This may enable the growth of more-tolerant pathogens, with potential clinical implications surrounding the initiation of supplemental oxygen and subsequent risk of CF exacerbations or disease progression. Further studies into the clinical effects of oxygen on the airway microbiome in pwCF, the mechanisms and timing of subsequent alterations, and potential interactions with other perturbations such as antibiotic use are warranted.

## MATERIALS AND METHODS

**Study inclusion.** Adult pwCF aged 18 years and older with a prior diagnosis of cystic fibrosis based on diagnostic criteria receiving routine care at the Massachusetts General Hospital Adult Cystic Fibrosis Center were recruited for a prospective cohort study. The study was given ethical approval by the Institutional Review Board of Mass General Brigham (protocol no. 2018P002934). Written informed consent was obtained from all participants. During routine outpatient clinical visits, expectorated sputum samples were obtained from adult pwCF and inoculated into culture on the same day. Clinical data were abstracted from the electronic medical record.

**Culture of airway microbial communities under different oxygen conditions.** Sputum samples were cultured under different oxygen conditions in artificial sputum medium on the day of sample collection (see Fig. 1 for a study overview). For each sputum sample, three autoclave-sterilized 500-mL serum bottles were prepared, one for each oxygen condition (21%, 50%, and 100%). Each serum bottle is inoculated with 24 mL of artificial sputum medium and 1 mL of patient sputum, sealed, and prepared for oxygen sparging (the process of replacing the internal atmosphere of the bottles with the desired oxygen concentration). Gas flow was set up to allow each bottle to undergo 10 air exchanges, and then the bottles were pressurized to +1 atmosphere. For each sputum sample inoculated into three serum bottles, one bottle was sparged to 21% oxygen, one to 50%, and one to 100%, representative of normal oxygen, moderate supplemental oxygen, and high supplemental oxygen conditions, respectively. Cultures were incubated at 37°C with orbital shaking at 150 rpm. At 24, 48, and 72 h after sputum inoculation, aliquots of cultured sputum were taken from each serum bottle and stored at −80°C until nucleic acid extraction. After aliquot removal, each serum bottle was resparged to the target oxygen concentration and reincubated until a total of 72 h of incubation time had elapsed. Figure S3 provides details on the observed microbes that were and were not able to be cultured, as well as the oxygen tolerance capability and Gram stain result for each species. Full details on the validation of this culture approach, including justification for the composition of artificial sputum medium and oxygen sparging protocol, have been previously described (27).

**Nucleic acid extraction and sequencing.** All samples were extracted and sequenced in the same batch prior to nucleic acid extraction, and 10 million cells of *Imtechella halotolerans*, a novel bacterium isolated from estuarine water and not found in human biological samples (55), were spiked into each sample (Zymo Research) for subsequent calculation of absolute microbial abundance. While a variety of methods exist for absolute abundance estimation (56), use of spike-in bacteria at the time of nucleic acid extraction is an approach that has been previously validated against other quantification methods of absolute abundance quantification (29).

The average sputum sample from pwCF is estimated to contain roughly 5 billion CFU per 0.5 mL sputum (our extraction volume), but the observed range varies greatly (58). Values as low as 100 million

CFU per milliliter were common, and thus, 10 million cells of *Imtechella halotolerans* (10 $\mu$L) was used to give an optimal working range between 100 million and 10 billion cells in a sample.

Samples, reagent-only negative controls, and mock community-positive controls (Zymo Research) were extracted using a protocol optimized for respiratory samples with a magnetic bead-based protocol using the Maxwell HT 96 genomic DNA (gDNA) blood isolation system (Promega) on a KingFisher Flex instrument as previously described (63). Briefly, 500 $\mu$L of each sputum sample and 500 $\mu$L of cetyl tri-methyl ammonium bromide (CTAB) were added to individual lysing matrix E tubes (MP Biomedicals), and the tubes were incubated at 95°C for 5 min, followed by bead beating for three 30-s cycles at 7.0 m/s, and then incubated with proteinase K at 70°C for 10 min. Then, 300-$\mu$L lysate samples were collected, followed by additional bead beating for three 30-s cycles at 7.0 m/s with each cycle, and additional 300-$\mu$L lysate samples were collected. Lysate samples were transferred to 96-well plates for binding, washing, and elution steps on the Kingfisher Flex sample purification system. Extracted nucleic acids were quantified using the PicoGreen double-stranded DNA (dsDNA) assay kit (Invitrogen), library preparation was performed using the Nextera XT DNA library preparation Kit (Illumina), and the library was sequenced on the NovaSeq 6000 platform to generate $2 \times 150$ base pair reads.

**Bioinformatics processing.** Raw data files in binary base call (BCL) format were converted into FastQ files and demultiplexed based on the dual-index barcodes using the Illumina "bcl2fastq" software. Whole-metagenome shotgun sequencing data were subsequently processed using bioBakery3 (28) version v3.0.0-alpha.6 (7-10-2020). Demultiplexed raw FastQ sequences were processed using KneadData (64), including the removal of human "contaminant" sequences, low-complexity and repetitive sequences, and adapter and low-quality bases with Trimmomatic (65), and contaminant checks were done with bowtie2 (66). For removal of spike-in bacterial reads, a MultiFasta file was constructed using downloaded reference genomes for the spike-in bacteria and used as the reference.

Taxonomic profiling of the sequenced samples at the species level was performed using MetaPhlAn3. Processed FastQ reads were first mapped against the MetaPhlAn3 (28) marker gene database (mpa_v30_CHOCOPhlAn_201901) to generate taxonomic profiles per sample. The output for all samples is a single taxon by sample table with estimated read counts and relative abundances. Functional profiling of the microbial community was performed using HUMAnN3 (28) and binned to the BioCyc (67) pathway database. Default pathway abundance and coverage tables, as well as gene family abundance output files per sample, were generated. All tables are split into stratified tables (by taxon) and unstratified (metagenome) tables.

In the case of HUMAnN3-predicted pathways, it was necessary to curate the large number of predicted functions to a narrow subset for focused analysis and visualization. Pathways were assigned to categories based on their BioCyc superclasses. With the added insight of this categorization, a subset of nonredundant pathways with activity related to oxygen were chosen. Particular focus was placed on respiration and fermentation reactions, as well as central metabolism, electron transport, stress signaling and antigen production, antibiotic resistance and production, and biosynthesis/breakdown reactions affected by oxygen (Table S4).

Antimicrobial resistance gene marker gene sequences were obtained from the Comprehensive Antibiotic Resistance Database (CARD) (68) version 3.0.7, and mobile genetic element sequences were obtained using a curated database (69) derived from the NCBI nucleotide database (70) and the PlasmidFinder database (71). Antimicrobial resistance profiles and mobile genetic element profiles were then generated for each sample with ShortBRED (72), using these databases as the references.

**Statistical analyses.** The complete R (https://www.R-project.org/) script used to analyze these data and generate the associated visualizations is attached as a supplemental document. Microbial features (microbial taxonomy, predicted function, AMR, and MGE profiles) and sample metadata were aggregated into phyloseq (73) for analysis. To calculate absolute abundance from taxonomic profiles, the signal attributed to the 10 million cells of *Imtechella halotolerans* spike-in bacteria was divided by its relative abundance, and then the 10 million cells was subtracted to yield the total estimated number of non-spike-in microbial cells for each sample:

$$\text{Cells}_{\text{sample}} = \frac{\text{Cells}_{\text{spike}}}{\text{RelAbund}_{\text{spike}}} - \text{Cells}_{\text{spike}}$$

$\text{Cells}_{\text{sample}}$ denotes the absolute abundance estimate for the cell count in the sample. $\text{Cells}_{\text{spike}}$ is the number of spike-in cells added. $\text{RelAbund}_{\text{spike}}$ is the resulting relative abundance of the spike-in bacteria with a range between 0 and 1.

From here, the sample absolute abundance estimate was multiplied by the relative abundance estimate of each taxon to yield per-taxon absolute abundance estimates:

$$\text{Cells}_{\text{taxon}} = \text{Cells}_{\text{sample}} \times \text{RelAbund}_{\text{taxon}}$$

$\text{Cells}_{\text{taxon}}$ is the absolute abundance estimate for the number of cells of a given taxon. $\text{Cells}_{\text{sample}}$ is the absolute abundance estimate for the whole sample. $\text{RelAbund}_{\text{taxon}}$ is the relative abundance of a given taxon with a range between 0 and 1.

To filter out potentially spurious features due to sequencing or classification errors, we performed prevalence filtering, excluding taxa and predicted functional pathways identified in less than 10% of samples (74). We did not perform abundance filtering, that is, rare taxa or pathways were retained if they were present in at least 10% of sequenced samples. Alpha and beta diversity metrics were calculated using phyloseq (73) and vegan (75).

To test the hypothesis that supplemental oxygen alters alpha diversity, we used linear mixed-effects

models as implemented in lmerTest (76). To test the hypothesis that supplemental oxygen alters microbial community structure, we performed permutational analysis of variance as implemented in vegan (75). Vegan was also used to calculate rarefied richness estimates. Differential abundance testing of microbes and pathways was performed with boosted additive general linear models as implemented in MaAsLin2 (77) accounting for repeated measures and multiple hypothesis. Results for differential abundance, as well as microbial metadata obtained from the BacDive (78) and BugBase (79) databases, were plotted using the Interactive Tree of Life (iTOL) (80).

Associations between microbes and functional pathways were determined using normalized Spearman's correlation and hierarchically clustered (81) as implemented by heatmaply (82). The microbial association network was calculated with NetCoMi (83) Spearman's correlation of absolute abundance estimates, carried out to 1,000 permutations. The visualized relationships were determined via Bonferroni-Hochberg corrected multiple-hypothesis testing with a $q$ value (false discovery rate) threshold of 0.1.

For additional details on the computational approaches applied to these data, please refer to the R analysis code document in the supplemental material.

**Data availability.** High-quality paired-end sequence and associated sample metadata were uploaded to the NCBI Sequence Read Archive repository under accession number PRJNA861321.

## SUPPLEMENTAL MATERIAL

Supplemental material is available online only.

**FIG S1**, TIF file, 2.8 MB.
**FIG S2**, TIF file, 0.1 MB.
**FIG S3**, TIF file, 1.6 MB.
**FIG S4**, TIF file, 0.2 MB.
**FIG S5**, TIF file, 0.1 MB.
**TABLE S1**, PDF file, 0.2 MB.
**TABLE S2**, PDF file, 0.2 MB.
**TABLE S3**, PDF file, 0.2 MB.
**TABLE S4**, PDF file, 0.3 MB.
**TABLE S5**, PDF file, 0.2 MB.

## ACKNOWLEDGMENTS

Funding was provided by National Institutes of Health grants K23 ES023700 and the Massachusetts General Hospital Department of Medicine Transformative Scholars Award. The funding agencies had no role in the design and conduct of the study; collection, management, analysis, and interpretation of the data; preparation, review, or approval of the manuscript; and decision to submit the manuscript for publication.

We declare we have no actual or potential competing financial interests.

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
