## [Reviewer comments · mSystems]

Supplemental oxygen alters the airway microbiome in cystic fibrosis

Jacob Vieira, Sirus Jesudasen, Lindsay Bringhurst, Hui-Yu Sui, Lauren McIver, Katrine Whiteson, Kurt Hanselmann, George O'Toole, Christopher Richards, Leonard Sicilian, Isabel Neuringer, and Peggy Lai

Corresponding Author(s): Peggy Lai, Massachusetts General Hospital

Review Timeline:

Submission Date:	April 21, 2022
Editorial Decision:	July 14, 2022
Revision Received:	July 25, 2022
Accepted:	July 29, 2022

Editor: Barbara Methe

Reviewer(s): The reviewers have opted to remain anonymous.

Transaction Report:

DOI: <https://doi.org/10.1128/msystems.00364-22>

July 14, 2022

Dr. Peggy S Lai
Massachusetts General Hospital
Pulmonary and Critical Care Medicine
Bulfinch 148
55 Fruit Street
Boston, MA 02114

Re: mSystems00364-22 (Supplemental oxygen alters the airway microbiome in cystic fibrosis)

Dear Dr. Peggy S Lai:

Thank you for submitting your manuscript to mSystems. We have completed our review and I am pleased to inform you that, in principle, we expect to accept it for publication in mSystems. However, acceptance will not be final until you have adequately addressed the reviewer comments.

Please address the reviewer comments especially Reviewer 4. Also, in the Discussion section in the strengths of the study paragraph beginning at line 364 "We evaluated changes in absolute rather than relative abundance..." This sentence should either be removed or you should amend it to include a statement that in this study you did not explicitly evaluate the accuracy of the spike-in used. Finally, please provide an update as to the public availability of the sequence data. Currently, there is only a statement "Experimental data will be made publicly available upon publication and upon request for peer review."

Preparing Revision Guidelines

Sincerely,

Barbara Methe

Editor, mSystems

Journals Department
Reviewer comments:

Reviewer #2 (Comments for the Author):

Vieira et al. have revised their manuscript and I have reviewed the new version and response to initial comments. Overall, I believe that the authors generally avoided addressing my main critiques with long and cumbersome responses often not relevant to the questions at hand, but buried within those responses, are some attempts to resolve the issues with their experimental design.

The principle issue of the lack of a completely anaerobic control was addressed with the argument that it is not needed and not relevant to the *in vivo* condition. This seems very counterintuitive to an argument they initially provide that there are steep oxygen gradients present in CF lungs, this I agree with, as it has been demonstrated before. Within these steep gradients are regions of anoxia (as the authors say in their response within 1 mm), therefore, it is difficult to reconcile the argument that "...purely anaerobic conditions would not replicate *in vivo* conditions." If the former argument is true.

Nevertheless, the authors do make the point that this study was more focused on 'hyperoxia'. In this context, the anaerobic control becomes less important, but still a concerning omission.

Furthermore, in supplemental figure 3 the authors show that true 'anaerobes' are present in their cultures, thus, the possibility that they were completely killed by the presence of oxygen is less concerning.

In sum, I believe that the logic behind the response to this critique is quite flawed and the tangential nature of their response overall is confusing, but with the manuscript's focus on hyperoxia, this issue is minimized.

It is peculiar that the authors then go on to argue why they did not use the WinCF model for their experiments. This was never suggested after my first review, the model itself well designed and logical. There is no 'gold standard' model for these types of experiments and their approach was fine. Their logic that WinCF could not be used for metagenomics only amplicon sequencing is also peculiar and not relevant to the critique provided.

The responses to my other critiques are satisfactory and I applaud the authors for their rigor in revising the manuscript accordingly.

Reviewer #3 (Comments for the Author):

The authors have responded well to the prior comments. This manuscript is clearly written, and the experiments well designed. The reported findings of changes in CF sputum microbiota with oxygen supplementation are a contribution to the field.

Reviewer #4 (Comments for the Author):

The authors present an *ex vivo* study of microbes from people with CF (pwCF) cultured under normoxia or two degrees of hyperoxia. I found the manuscript well-written and thorough, with appropriately drawn conclusions. The study design also seems appropriate, especially given the rationale the authors provided for focusing on the difference between normoxia and hyperoxia specifically. In terms of methods, while the study is admittedly small, I think the authors' use of metagenomics with spike-ins to quantify microbial load already puts them ahead of the majority of studies in the field, which only attempt to quantify relative abundance. Most of my comments are fairly minor:

- It looks like there's a missing word or phrase in the abstract (p. 3, line 62).
- The section on co-occurrence networks (p.13) is difficult to interpret. To what extent are these metrics capturing something besides the overall trend of oxygen reducing the complexity of the community? And which metrics are the most critical to discuss here? Because these networks are calculated based on pairwise Spearman correlations and because the overall number of sputum samples is low, I also wonder how much these summaries might be driven by strong trends "within" a minority of sputum samples, as opposed to trends that are broadly reproduced across samples.
- I found the sentence on line 257 ("This steep drop-off in mediating relationships could eliminate associations that help control

the abundance of pathogens like *Pseudomonas* and *Staphylococcus* under normoxia") to be a little speculative, at least in the absence of more supporting details.

- I was surprised based on Figure 2A-B that absolute microbial load had significant oxygen and especially time trends. The effects seem much stronger for the other variables. Is this a case of the box plot failing to capture something important in the data, or is something else going on? (The Shannon Index plots seem to have a much more convincing time trend but the effect size and CI are similar, for instance.)

- The x-tick labels in Figure 3AB appear to be misaligned.

- Did read depth correlate with absolute microbial load? This is obviously not necessarily true, but a correlation has been previously noted in the gut microbiome literature.

Dear Dr. Methe,

We thank the reviewers and editor for the time spent on reviewing our manuscripts and the thoughtful comments provided. We would like to address comments in a point-by-point response below.

EDITOR:

Please address the reviewer comments especially Reviewer 4.

Response: This has been done as suggested, see below point-by-point response.

Also, in the Discussion section in the strengths of the study paragraph beginning at line 364 "We evaluated changes in absolute rather than relative abundance..." This sentence should either be removed or you should amend it to include a statement that in this study you did not explicitly evaluate the accuracy of the spike-in used.

Response: We have revised this sentence as follows:

“We evaluated changes in absolute rather than relative abundance in our taxonomic profiles with the use of spike-in controls prior to nucleic acid extraction although we did not verify the derived absolute abundance estimates.” (line 388)

Finally, please provide an update as to the public availability of the sequence data. Currently, there is only a statement "Experimental data will be made publicly available upon publication and upon request for peer review."

Response: In the revised manuscript we provide the NCBI accession number for these data as follows in the text:

“High-quality paired-end sequence and associated sample metadata were uploaded to the NCBI Sequence Read Archive repository under accession number PRJNA861321.” (line 578)

REVIEWER 2:

Vieira et al. have revised their manuscript and I have reviewed the new version and response to initial comments. Overall, I believe that the authors generally avoided addressing my main critiques with long and cumbersome responses often not relevant to the questions at hand, but buried within those responses, are some attempts to resolve the issues with their experimental design.

The principle issue of the lack of a completely anaerobic control was addressed with the argument that it is not needed and not relevant to the in vivo condition. This seems very counterintuitive to an argument they initially provide that there are steep oxygen gradients present in CF lungs, this I agree with, as it has been demonstrated before. Within these

steep gradients are regions of anoxia (as the authors say in their response within 1 mm), therefore, it is difficult to reconcile the argument that "...purely anaerobic conditions would not replicate in vivo conditions." If the former argument is true.

Nevertheless, the authors do make the point that this study was more focused on 'hyperoxia'. In this context, the anaerobic control becomes less important, but still a concerning omission.

Furthermore, in supplemental figure 3 the authors show that true 'anaerobes' are present in their cultures, thus, the possibility that they were completely killed by the presence of oxygen is less concerning.

In sum, I believe that the logic behind the response to this critique is quite flawed and the tangential nature of their response overall is confusing, but with the manuscript's focus on hyperoxia, this issue is minimized.

It is peculiar that the authors then go on to argue why they did not use the WinCF model for their experiments. This was never suggested after my first review, the model itself well designed and logical. There is no 'gold standard' model for these types of experiments and their approach was fine. Their logic that WinCF could not be used for metagenomics only amplicon sequencing is also peculiar and not relevant to the critique provided.

Response: We thank the reviewer for the time taken to critique this manuscript. We have further emphasized the lack of an anaerobic culture condition as a limitation of the study in the discussion:

“In this model system, we did not include a purely anaerobic condition, which is a limitation, although we did still detect anaerobes in our 21% oxygen culture condition.”
(line 411)

Regarding the WinCF model, we were concerned that the small culture volume in the capillary tubes would lead to insufficient biomass for metagenomics sequencing, although as the reviewer notes there are ways to optimize low input samples for metagenomics sequencing.

The responses to my other critiques are satisfactory and I applaud the authors for their rigor in revising the manuscript accordingly.

REVIEWER 3:

The authors have responded well to the prior comments. This manuscript is clearly written, and the experiments well designed. The reported findings of changes in CF sputum microbiota with oxygen supplementation are a contribution to the field.

Response: We thank the reviewer for taking the time to review the manuscript, critiques, and responses.

REVIEWER 4:

The authors present an *ex vivo* study of microbes from people with CF (pwCF) cultured under normoxia or two degrees of hyperoxia. I found the manuscript well-written and thorough, with appropriately drawn conclusions. The study design also seems appropriate, especially given the rationale the authors provided for focusing on the difference between normoxia and hyperoxia specifically. In terms of methods, while the study is admittedly small, I think the authors' use of metagenomics with spike-ins to quantify microbial load already puts them ahead of the majority of studies in the field, which only attempt to quantify relative abundance. Most of my comments are fairly minor:

It looks like there's a missing word or phrase in the abstract (p. 3, line 62).

Response: We have corrected the abstract so it now reads as follows:

“Hyperoxia reduced absolute abundance of specific microbes including facultative anaerobes such as *Rothia* and some *Streptococcus* species, with minimal impact on canonical CF pathogens such as *Pseudomonas aeruginosa* and *Staphylococcus aureus*.”
(line 62)

The section on co-occurrence networks (p.13) is difficult to interpret. To what extent are these metrics capturing something besides the overall trend of oxygen reducing the complexity of the community? And which metrics are the most critical to discuss here?

Response: The purpose of the network analysis was to understand how hyperoxia perturbs airway microbial communities beyond simple measures of reduced alpha diversity. There is a presumption that higher diversity equates a more stable microbial network but this is not true in all contexts; work by Coyte et al. ¹ has shown that microbial networks without competition (here we attempt to approximate competitive (mediating) interactions with negative edge percentage) may be less stable. Thus, we felt that beyond alpha diversity summaries, it was important to evaluate the effect of hyperoxia on the microbial community using network statistics as well. This rationale is highlighted in the results as follows:

“Although it is often assumed that microbial communities with higher diversity are also more stable, this is not always the case as ecological models indicate that competitive relationships may stabilize microbial networks(32). Thus, we evaluated the effect of hyperoxia on microbial co-occurrence networks (**Figure 6**) and compared network statistics on communities cultured under 21% and 100% oxygen culture samples.” (line 248)

For microbial network analysis, as the reviewer points out, there is less agreement about which network metrics to report and few available tools to facilitate statistical comparisons between networks suitable for microbial community data. Our goal was to report network metrics showing that (1) Hyperoxia led to statistically significant differences in overall network topology thus the choice to report the Adjusted Rand Index and comparisons of centrality and connectedness measures between the microbial communities cultured under 21% vs 100% oxygen; (2) Hyperoxia fragmented the microbial network, thus the choice to report the number

of network components; (3) Hyperoxia may have reduced competition, thus the choice to report the positive (which imply synergistic (cooperative)) as opposed to negative (mediating (competitive)) relationships between microbes.

To better highlight the rationale for the choice of network statistics, we have modified the results as follows and now report the negative edge percentage rather than the positive edge percentage:

“Exposure to a hyperoxic environment leads to global changes in network topology (**Supplemental Table 5**). Comparing the overall similarity of the two networks yields an Adjusted Rand Index of 0.462 ($p < 0.001$), indicating only 46.2% agreement in microbial pair placement between the two sets. There is 92% dissimilarity between global degree centrality ($p = 0.004$) and a shift in network density from 0.308 to 0.150 ($p = 0.068$) under 100% oxygen conditions. Hyperoxia lead to fragmentation of the microbial network. While the normoxic microbial association network is unified into a single component, the hyperoxic network is broken into 16 components ($p = 0.001$), 12 of which are singlets isolated by the strong depletion of that species’ presence under hyperoxia. These 3 metrics point to a significant overall sparsification of microbial associations under hyperoxic conditions. Within the remaining sparser network under hyperoxia, the cluster coefficient increases from 0.688 to 0.841 ($p = 0.002$), indicating tighter cluster formation among the remaining relationships. Hyperoxia may have reduced competition, approximated by the percentage of negative edges (negative correlations) between microbial species. The overall negative edge percentage decreased from 12.1% to 2.3% (21% vs 100% oxygen, $p = 0.004$), indicating a depletion of significant mediating (competitive) relationships.“ (line 252)

Because these networks are calculated based on pairwise Spearman correlations and because the overall number of sputum samples is low, I also wonder how much these summaries might be driven by strong trends "within" a minority of sputum samples, as opposed to trends that are broadly reproduced across samples.

Response: Airway microbiome studies in pwCF² may differ from other human microbiome studies in the very strong subject-specific effects that exist; i.e. the airway microbiome of each pwCF is unique (as depicted in **Figure 3**). In a sense every person is an “outlier” since there is no “average” community composition of the airway microbiome in pwCF, so we do not believe our results are driven by a few outliers though we do highlight as a limitation of this study the smaller sample size (number of patients).

I found the sentence on line 257 ("This steep drop-off in mediating relationships could eliminate associations that help control the abundance of pathogens like Pseudomonas and Staphylococcus under normoxia") to be a little speculative, at least in the absence of more supporting details.

Response: We have removed this sentence as suggested.

I was surprised based on Figure 2A-B that absolute microbial load had significant oxygen and especially time trends. The effects seem much stronger for the other variables. Is this a case of the box plot failing to capture something important in the data, or is something else

going on? (The Shannon Index plots seem to have a much more convincing time trend but the effect size and CI are similar, for instance.)

Response: Absolute microbial load estimates vary between samples across a couple orders of magnitude. Thus in the boxplots, unlike the measures of alpha diversity, absolute microbial load was log-transformed on the y-axis of the box-plot, which can make clear parsing of signal more difficult between oxygen and time conditions. We suspect that is why differences across oxygen and time conditions are visually less obvious for absolute microbial load in the box plot.

The x-tick labels in Figure 3AB appear to be misaligned.

Response: We modified the text rotation code producing these figures to better align the tick labels. This was done in this figure and other figures with the same issue.

Did read depth correlate with absolute microbial load? This is obviously not necessarily true, but a correlation has been previously noted in the gut microbiome literature.

Response: Estimated microbial load did have a modest correlation with read depth, with a correlation coefficient of 0.54. We calibrated the amount of spike-in microbes added to each sample so that the final reads aligned to the spike-in was within the range of 0.1 – 10% of final quality and host filtered reads, and our average sequencing depth was 23.3 million reads per sample, thus any differences in sequencing depth were unlikely to have affected our calculations of absolute abundance using spike-ins.

References cited

1. Coyte KZ, Schluter J, Foster KR. The ecology of the microbiome: Networks, competition, and stability. *Science*. 2015;350(6261):663-6. doi: 10.1126/science.aad2602. PubMed PMID: 26542567.
2. Comstock WJ, Huh E, Weekes R, Watson C, Xu T, Dorrestein PC, Quinn RA. The WinCF Model - An Inexpensive and Tractable Microcosm of a Mucus Plugged Bronchiole to Study the Microbiology of Lung Infections. *J Vis Exp*. 2017(123). Epub 2017/05/19. doi: 10.3791/55532. PubMed PMID: 28518116.

July 29, 2022

Dr. Peggy S Lai
Massachusetts General Hospital
Pulmonary and Critical Care Medicine
Bulfinch 148
55 Fruit Street
Boston, MA 02114

Re: mSystems00364-22R1 (Supplemental oxygen alters the airway microbiome in cystic fibrosis)

Dear Dr. Peggy S Lai:

Your manuscript has been accepted, and I am forwarding it to the ASM Journals Department for publication. For your reference, ASM Journals' address is given below. Before it can be scheduled for publication, your manuscript will be checked by the mSystems production staff to make sure that all elements meet the technical requirements for publication. They will contact you if anything needs to be revised before copyediting and production can begin. Otherwise, you will be notified when your proofs are ready to be viewed.

Publication Fees:

If you would like to submit a potential Featured Image, please email a file and a short legend to mSystems@asmusa.org. Please note that we can only consider images that (i) the authors created or own and (ii) have not been previously published. By submitting, you agree that the image can be used under the same terms as the published article. File requirements: square dimensions (4" x 4"), 300 dpi resolution, RGB colorspace, TIF file format.

We recognize that the video files can become quite large, and so to avoid quality loss ASM suggests sending the video file via <https://www.wetransfer.com/>. When you have a final version of the video and the still ready to share, please send it to mSystems staff at mSystems@asmusa.org.

Sincerely,

Barbara Methe
Editor, mSystems

Journals Department
Figure S3: Accept
Supplemental Material: Accept
Table S1: Accept
Table S5: Accept
Figure S1: Accept
Table S2: Accept
Figure S2: Accept
Figure S5: Accept
Table S4: Accept
Figure S4: Accept